# Characterization of Differences in the Composition and Content of Volatile Compounds in Cucumber Fruit

**DOI:** 10.3390/foods11081101

**Published:** 2022-04-12

**Authors:** Jie Zhang, Xiuchao Gu, Wenjing Yan, Lina Lou, Xuewen Xu, Xuehao Chen

**Affiliations:** 1School of Horticulture and Plant Protection, Yangzhou University, Yangzhou 225009, China; zj20220211@163.com (J.Z.); gxc172562666@gmail.com (X.G.); wenjingyan0024@163.com (W.Y.); xxu323@yzu.edu.cn (X.C.); 2Jiangsu Key Laboratory for Horticultural Crop Genetic Improvement, Institute of Vegetable Crops, Jiangsu Academy of Agricultural Sciences, Nanjing 210014, China; linabeibei@163.com; 3Joint International Research Laboratory of Agriculture and Agri-Product Safety, The Ministry of Education of China, Yangzhou University, Yangzhou 225009, China

**Keywords:** cucumber, volatile organic compounds, flavor, HS-SPME/GC–MS

## Abstract

The cucumber is characterized by the presence of a wide range of volatile organic compounds (VOCs), which are recognized as the main responsible for its unique flavor. However, research on the types and contents of VOCs in different cucumber cultivars remains fragmentary. Here, using an automatic headspace solid-phase microextraction coupled with the gas chromatography–mass spectrometry method, the VOCs were analyzed in three representative cucumber cultivars, including YX, KX, and GX, with the best, middle, and worst flavor quality, respectively, which were selected from 30 cultivars after flavor quality evaluation. Principal component analysis revealed that the six biological replicates were grouped, indicating high reliability of the data. A total of 163 VOCs were detected. There were 28 differential VOCs in YX compared to GX, 33 differential VOCs in YX compared to KX, and 10 differential VOCs in KX compared to GX. Furthermore, K-means clustering analysis showed that 38 of the 43 no-overlapping differential VOCs were represented by the most abundant compounds detected in YX. The prevailing VOCs in YX included: hydrocarbons, aldehydes, and ketones. The data obtained in the present study extend our understanding the impact of cultivars on VOCs in cucumber and will help facilitate targeted breeding.

## 1. Introduction

Cucumber (*Cucumis sativus* L., 2n = 2X = 14) is widely cultivated and consumed all over the world [1]. The fruits of cucumbers are fragrant and delicious, with nutrient enrichment that can be consumed fresh, cooked, or pickled [2,3]. With the ever-rising living standards of people, the flavor quality of cucumbers is becoming much more concerned by the consumers [4]. Thus, a comprehensive understanding of fruit flavor-related metabolites and their improvement should be included in the breeding objective [5].

VOCs contribute to fruit flavor [6]. According to their biosynthetic sources, VOCs can be roughly divided into the following three categories: (1) terpenoids generated by methylerythritol phosphate (MEP) or mevalanoic acid (MVA); (2) phenylpropanoids or benzenes derived from aromatic amino acids; (3) alcohols or aldehydes derived from unsaturated fatty acids and amino acids [7]. According to the synthesis and release sites, VOCs can be divided according to their production by either reproductive organs or vegetative organs. During plant domestication, the synthesis of VOCs by fruit can improve its taste and attract predators, resulting in the seeds becoming widely spread [8]. In daily life, VOCs improve the flavor of fruits, which can stimulate consumers’ desire to purchase them. In addition, VOCs have antibacterial efficacy, which can help to prolong the fruit storage life [9]. Oliveira et al. [10] showed that the peel, pulp, and leaves from different fig (*Ficus carica*) cultivars can be distinguished by their distinct abundance of monoterpenes, sesquiterpenes, and aldehydes. Through the correlation analysis between the VOCs and the sensory descriptors, de Freitas et al. [11] found that the methyl butanoate, methyl 3-methylbutanoate, ethyl 2-methylbutanoate, methyl 2-butenoate, methyl 3-methylpentanoate, 3-carene, methyl (*E*)-2-methyl-2-butenoate, ethyl 4-methylpentanoate, 2-hexenal, butyl 3-methylbutanoate, butyl pentanoate, and 3-methyl butanoic acid were key factors in explaining differences in the characteristic fruit aroma and flavor of cashew apples.

In melon, at least 500 VOCs have been identified and more than 100 of them can be found in a single accession [12]. Recently, Mayobre et al. [13] found that the non-aromatic melon Piel de Sapo and the aromatic melon Védrantais produced 88 different VOCs. For the first time, Guler et al. [14] found that (*E*,*Z*)-2,4-heptadienal, (*E*,*E*)-3,5-octadien-2-one, and 3-decyne were cucumber VOCs. Wei et al. [15] qualitatively and quantitatively analyzed 85 volatile chemicals, including 36 volatile terpenes, in 23 tissues of cucumber plants using solid-phase microextraction (SPME) combined with gas chromatography–mass spectrometry (GC–MS). However, research on the types and content of VOCs in different cucumber cultivars is still lacking. To investigate differences in the composition and content of VOCs in cucumber fruits, three representative cultivars with distinct flavors were selected as test materials following the flavor-quality evaluation of 30 cultivars, and these were European greenhouse cucumber Yuxiu2 (YX), Northern Chinese-type cucumber Kangxiu4 (KX), and American pickling-type cucumber GX312 (GX). Automatic headspace SPME combined with GC–MS (HS-SPME/GC–MS) was used to analyze and compare the VOCs in the fruits of the three cucumber cultivars. The aim of this work is to study the impact of cultivars on the concentration and combination of VOCs in cucumbers, which could be integrated into breeding programs for future flavor improvement.

## 2. Materials and Methods

### 2.1. Sample Preparation

The 30 test cucumber cultivars were provided by the Cucumber Heritage Breeding Group of the College of Horticulture and Plant Protection, Yangzhou University (Yangzhou, China). Twenty plants of each cultivar were grown in a greenhouse. The management conditions were consistent throughout the cultivation process. Twelve days after flowering, 10 well-shaped fruits among 5 to 12 nodes were selected from each variety. They were immediately brought back to the laboratory after picking. Among each variety, four fruits were used for VOC identification and analysis. The cucumbers were cut in half lengthwise, and the peels and seeds were also removed. Then, the flesh of each fruit was carefully chopped, and each flesh sample was wrapped in foil weighing about 30 g. After each sample was taken, it was immediately stored at −80 °C. The remaining six fruits of each variety were immediately stored at 4 °C for subsequent flavor quality evaluation.

### 2.2. Flavor Quality Evaluation

Sensory evaluation was conducted by 10 trained graduate students with five men and five women, aged 21–25. The panelists were all from the School of Horticulture and Plant Protection, Yangzhou University (Yangzhou, China). They were in good physical condition (no smell or taste disorders) and had received relevant training. The rating was based on a clear flavor, sweetness, and astringency on a scale of 0–9 (0 being odorless and 9 being the strongest). The fresh fruit of each cucumber variety was cut into 2 cm thick slices, and the varieties were randomly placed. During the evaluation, the panelists were not allowed to communicate with each other and gargled water immediately after tasting a variety to reduce experimental errors. The whole evaluation process was conducted at room temperature, and all participants signed informed consent.

Total soluble solids (TSS) content was measured using a digital refractometer (Model PAL-1, ATAGO, Tokyo, Japan). Before testing, water was poured into the sample tank to zero the instrument, in which the screen displayed 0.0% to indicate that the calibration was successful. Then, the juice from the flesh of each variety was forcefully extruded and dropped into the sample tank (about 500 μL), and the TSS content was immediately determined and recorded.

### 2.3. Instruments and Chemical Reagents

For gas chromatography–mass spectrometry (model 8890-5977B), we used a chromatographic column (model DB-5MS, 30 m × 0.25 mm × 0.25 μm) and extraction fiber (120 μm DVB/CAR/PDMS), which were manufactured by Agilent, Santa Clara, CA, USA. The ball mill (model MM400) was manufactured by Retsch, Haan, Germany. The electronic balance (model MS105DU) was a product of METTLER TOLEDO, Switzerland. The solid-phase microextraction unit, aging device, and sample heating chamber were products of CTC Analytics AG, Zwingen, Switzerland. The thermostat water bath was a product of J. P. Selecta, Spain. Sodium chloride was from Sinopharm, China. n-Hexane was a product of Merck, Germany. The standard (stored at −20) was from BioBioPha/Sigma-Aldrich, Darmstadt, Germany.

### 2.4. SPME Sampling

The samples were removed from the −80 °C freezer and ground into a fine powder with liquid nitrogen. Six replicate samples were taken from each variety. Each sample weighed about 1 g and was placed in a headspace bottle containing 10 μL saturated sodium chloride solution and 10 μL 2-methyl 3-heptanone internal standard solution.

The headspace bottle was sealed and then placed in a 100 °C constant temperature water bath and shaken for 5 min. Then, the 120 µm DVB/CAR/PDMS extraction fiber was inserted into the sample headspace bottle (the extraction head should be heated and aged for 5 min in the aging device in advance) for 15 min to absorb VOCs. The fiber was conditioned prior to use by heating in the injection port of the GC apparatus (Model 8890; Agilent) at 250 °C for 5 min.

### 2.5. GC–MS Analysis

The chromatographic conditions were as follows: DB-5MS capillary column (30 m × 0.25 mm × 0.25 μm) as the chromatographic column, high purity helium (purity no less than 99.999%) as the carrier gas, constant flow rate of 1.0 mL/min, inlet temperature of 250 °C, nonfractional injection, and solvent delay of 3.5 min.

Heating procedure: 40 °C for 3.5 min, increase to 100 °C at 10 °C/min, increase to 180 °C at 7 °C/min, and increase to 280 °C at 25 °C/min, with the final temperature maintained for 5 min.

The mass spectrometry conditions were as follows: electron bombardment ion source (EI) as the ionization mode, ion source temperature of 230 °C, four-stage bar temperature of 150 ℃, mass spectrometry interface temperature of 280 °C, ionization voltage of 70 eV, and scanning in full-scan mode (scan mass range: *m*/*z* 50–500 amu).

Identification of volatile compounds was achieved by matching the mass spectra with the self-built MWGC database (Metware Biotechnology Co., Ltd. Wuhan, China) or the standard National Institute of Standards and Technology library and based on the linear retention indexes of compounds.

### 2.6. Statistical Analysis

To ensure the accuracy of the determined values, six replicates were analyzed for each cultivar. The original data files obtained by GC–MS analysis were extracted using MassHunter software (Agilent, USA), and the mass-to-charge ratio, retention time, and peak area of characteristic peaks were obtained and then analyzed statistically. Then, the original data retention index was calculated, single peaks were filtered, and quantitative analysis was carried out by an internal standard normalization method. The statistical function prcomp within R (www.r-project.org) (accessed on 4 September 2021) was used for unsupervised principal component analysis. The cluster analysis results of VOCs were presented as heatmaps. OPLS-DA was used to detect differences between groups. The PCA was carried out by the R base package (v. 3.5.0), clustering heatmaps were created using the R package pheatmap (v. 1.0.12), and OPLS-DA was carried out using the R package MetaboAnalystR (v. 1.0.1).

## 3. Results

### 3.1. Test Material Selection

A total of 30 cucumber cultivars were evaluated for their flavor quality using a sensory evaluation index combined with TSS detection. The results are presented in Table 1. In general, the tested cucumber cultivars showed variation in their fresh cucumber-like flavors, sweetness, astringency, and TSS content. Combined with each index, it was found that YX had the best flavor quality, KX was in the middle, and GX was the worst among the 30 evaluated cultivars. Therefore, these three cucumber cultivars with a large flavor discrepancy were selected for the following VOC analysis (Figure 1).

### 3.2. Identification and Quantitative Analysis of Volatile Compounds

A wide variety of VOCs were identified from the three selected cucumber varieties using HS-SPME/GC–MS analysis. Six replicate samples were taken from the flesh of three cucumber varieties. Details of the types and amounts of VOCs detected are provided in the Appendix A. A total of 163 VOCs were detected in the 18 cucumber flesh samples, including 7 amines, 14 alcohols, 6 aromatic compounds, 5 phenolics, 2 sulfur compounds, 8 halogenated hydrocarbons, 23 aldehydes, 1 acid, 8 terpenoid substances, 25 hydrocarbons, 12 ketones, 32 heterocyclic compounds, and 20 esters (Appendix A). These included (*E*,*Z*)-2,6-nonadienal, (*E*)-2-nonenal, 2,4-heptadienal, lauraldehyde, hexanal, and caryophyllene, among others. Of these, (*E*,*Z*)-2,6-nonadienal is the substance found to be responsible for conferring cucumbers their characteristic aroma [16].

YX had the highest total VOC content, followed by KX, while GX had the lowest. Moreover, the contents of these volatile compounds mainly related to fruit flavor were found to be higher in YX (Appendix A). Figure 2 shows the GC–MS total-ion current chromatograms of the VOCs in the flesh of the three cucumber cultivars.

### 3.3. Principal Component and Cluster Analysis

Principal component analysis (PCA) was used for preliminary classification of all samples to reveal the separate trends of each group. Through PCA of fresh samples of the three different cucumber cultivars, the overall metabolic differences among varieties and the degree of variation among samples within the group could be obtained. PC1 and PC2 explained 45.46% and 18.81% of the phenotypic differences, respectively. As can be seen from Figure 3, the overall VOCs of the three cucumber cultivars were significantly different, while there was little variation between the six replicates within each cultivar.

A cluster analysis was further performed. Prior to analysis, unit variance scaling (UV) was used to normalize all the VOC data, and a cluster heatmap was drawn using the R software pheatmap package to analyze the metabolite accumulation patterns of different cucumber cultivars (Figure 4). According to the cluster analysis heatmap, the 163 VOCs could be divided into 15 categories. On the whole, the three cucumber cultivars could be ranked according to VOC as YX > KX > GX. According to their contents, heterocyclic compounds were the most abundant of the VOCs, followed by hydrocarbons and aldehydes. The remaining compounds with less content were found according to the following decreasing order of content: esters, alcohol, terpenoids, aromatics, ketones, amine, others, phenols, halogenated hydrocarbons, acids, nitrogen compounds, and sulfur compounds. Only a few VOCs, such as aldehydes, can produce the characteristic smell of cucumber [17]. However, the roles of other VOCs cannot be ignored. Although their contents are low, they still play an auxiliary and harmonizing role in fruit flavor [18]. The contribution of each VOC depends on its respective odor threshold and interactions with other compounds [19].

### 3.4. Differential Metabolite Selection

To further identify the differential metabolites among different varieties, orthogonal partial least-squares discriminant analysis (OPLS-DA) was used. OPLS-DA can maximize the differentiation between groups and facilitate the search for differential metabolites [20]. The selection criteria were as follows: fold change ≥ 2 or ≤ 0.5, *p* < 0.05; variable importance in projection (VIP) ≥ 1. To avoid overfitting, a permutation test (200 permutations) was performed. When these criteria are met, it is considered that there is a significant difference in the considered metabolite. Meanwhile, the relationship between different metabolites in each group is shown in a Venn diagram (Figure 5).

A total of 28 differential metabolites were identified in YX and GX, of which 5 were upregulated and 23 downregulated, and 3 were unique (Appendix A). A total of 33 differential metabolites were identified in YX and KX, among which 5 were upregulated and 28 downregulated, and 15 were unique (Appendix A). A total of 10 differential metabolites were identified in KX and GX, all of which were downregulated and none were unique, i.e., they were identified in the other two varieties (Appendix A). Only three differential metabolites were identified in all three groups, and the highest contents were found in YX, which were alpha-cadinol (terpenoid), *N*-(2,3,4-trifluorobenzoyl)-l-alanine-methyl ester (ester), and [3S-(3-alpha,5-alpha,8-alpha)]-5-azulenemethanol,1,2,3,4,5,6,7,8-octahydro-alpha,alpha,3,8-tetramethyl-acetate (ester).

### 3.5. K-Means Clustering Analysis of Differential Metabolites

To explore the variation trend of relative contents of differential metabolites in different samples, the relative contents of differential metabolites were standardized and centralized. Then, K-means clustering analysis was performed (Figure 6), and the specific classifications and contents of differential metabolites are shown in Table 2.

It can be seen from Figure 6 that the 43 differential metabolites were divided into four categories, among which Subclasses 1, 3, and 4 (38 kinds of differential metabolites in total) represented the most abundant compounds detected in YX, while Subclass 2 (only five kinds of differential metabolites) compounds were those found to be the highest detected in GX. Table 2 shows that hydrocarbons, aldehydes, and ketones accounted for the largest proportions in Subclass 1, and a small number of esters, halogenated hydrocarbons, heterocyclic compounds, alcohols, and others were also included. Alcohol accounted for the most abundant of the Subclass 2 compounds, with a small number of heterocyclic compounds, terpenoids, and halogenated hydrocarbons also included. Aldehydes, hydrocarbons, and heterocyclic compounds accounted for the most abundant of Subclass 3 compounds, and a small number of terpenoids, esters, and amines were also found. Terpenoids and hydrocarbons accounted for the largest proportion of Subclass 4 compounds detected, while there were fewer amines and esters. Among the detected metabolites, 1-iodo-hexadecane was extremely scarce in YX and relatively abundant in KX and GX. Aldehydes, alcohols, and esters are important compounds closely related to melon aroma and taste [21,22], and the contents of these compounds in YX were significantly higher than those in KX and GX, which provides a rationale for the reason why fruit cucumbers are found to have a more appealing taste.

## 4. Discussion

Nowadays, people have increasingly higher requirements for food, so the flavor quality of fruits and vegetables has become a research hotspot, and VOCs are the factors significantly affecting flavor quality [23,24]. Most of these VOCs have a certain flavor, and their presence can effectively improve the flavor of fruits and vegetables and bestow high sensory and physiological value [25,26]. At the same time, the combination of plant VOCs and nonvolatile special chemicals can protect plants from various forms of biological attack and promote their adaptation to the environment [27]. In addition, VOCs released by plants can alter the ecological environment. For example, isoprene compounds can change the ozone concentrations in the atmosphere to some extent [28]. Therefore, VOCs play an important role in nature.

Sensory quality analysis is a scientific method that has been widely used in food evaluation and involves the perception of the sensory characteristics of products through vision, smell, and taste [29]. This method more accurately reflects the actual preferences of consumers; on the other hand, it is also susceptible to the subjective factors of evaluators, resulting in deviation in the results [30]. Thus, assistance from instrument detection, which is more accurate, objective, and reproducible, is required for the comprehensive evaluation of flavor quality [31]. In this study, the flavor quality of 30 cucumber cultivars was evaluated, and selected 3 cultivars with distinct flavors as representatives. The evaluation involved sensory evaluation in combination with TSS detection because previous studies have shown a positive correlation between the TSS content and cucumber flavor [32]. Moreover, a total of 163 VOCs were detected in the three cucumber cultivars, which consisted of 13 types of compounds, namely, amines, alcohols, aromatic hydrocarbons, phenols, sulfur-containing compounds, halogenated hydrocarbons, aldehydes, acids, terpenoids, hydrocarbons, ketones, esters, and heterocyclic compounds. Among these VOCs, the contents of (*E*,*Z*)-2,6-nonadienal and (*E*)-2-nonenal, which are considered to be the main flavor substances of cucumber [33], [34,35]; furans are oxygen-containing heterocyclic compounds with a fruity and fresh fragrance [36]; esters are the main type of aromatic compounds and determine the unique fragrance of many fruits and vegetables [37,38]; terpenoids are the main source of flower fragrance [39], with a pleasant aroma and improve the taste of fruit; acids can regulate fruit acidity, and fatty acids-derived VOCs make significant contributions to tomato fruit flavor and human preferences [40]. Here, in this study, the contents of these important VOCs were found to be higher in YX than in the other two cultivars. This leads us to propose that different varieties of cucumber contain different types and amounts of VOCs, resulting in their taste differences. However, it should be pointed out that the cucumber fruit samples were stored at −80 °C for 6 days before the HS-SPME/GC–MS analysis, which might have affected the volatile composition. In addition, during SPME sampling, the headspace bottles were placed at 100 °C for 5 min, which may have changed the chemical structures of cucumber fruits, resulting in altered VOCs. Despite GC is used to separate the volatile and thermally stable substitutes in a sample, further studies with other alternative testing methods are needed to eliminate the potential side effects of thermal processing.

It has been reported that there are similar volatile components present in Northern Chinese-type cucumber M13, Southern Chinese-type cucumber Quanzao, and European greenhouse cucumber SW, and the contents of furan compounds in VOCs detected in M13 and Quanzao are higher, while those in SW are relatively lower [41]. Moreover, the Northern Chinese-type cucumber Crete contains higher octadecatrienal and pentadecanol contents, while the European greenhouse cucumber Knossos45 contains higher contents of (*E*,*Z*)-2,6-nonadienal and (*E*)-2-nonenal, which gives the latter a more pleasant flavor [42]. In this study, the main flavor compounds, including (*E*,*Z*)-2,6-nonadienal, (*E*)-2-nonenal, and 2,4-heptadienal, were significantly higher in YX than in KX and GX, which was consistent with previous research conclusions. However, the contents of furans such as 2-n-butyl furan and 2,2′,5,5′-tetrahydro-2,2′-bifuran were higher in YX than in the two other cultivars, which may be related to the detection methods and varieties. In the analysis and comparison of differences between the three groups, only three VOCs were detected in all, namely, alpha-cadinol, *N*-(2,3,4-trifluorobenzoyl)-l-alanine-methyl ester, and [3*S*-(3-alpha,5-alpha,8-alpha)]-5-azulenemethanol,1,2,3,4,5,6,7,8-octahydro-alpha,alpha,3,8-tetramethyl-acetate. These VOCs have not previously been reported, and how they interact and their metabolic pathways require further study.

This study showed that significantly different VOCs were detected in the three selected cucumber cultivars. This is mainly because the genotype can affect the VOC content of a fruit, so that different cultivars have different tastes [4]. However, research into the molecular mechanisms of cucumber VOC biosynthesis lags behind that of nonvolatile compounds [43]. Existing studies have identified representative candidate genes that are likely involved in the production of terpenoids, benzenoids, and C6/C9 aldehydes/alcohols in cucumber. Moreover, *TPS11*/*TPS14*, *TPS01*, and *TPS15* may be involved in the synthesis of terpenoids in the roots, flowers, and fruits of cucumber, respectively. Studies have also found that *SSUI* may regulate the synthesis of the volatile monoterpenoid precursor geranyl diphosphate in roots and flowers [7,27,44], whereas the molecular basis of other important VOCs remains to be explored.

In addition to genetic mechanisms, the VOCs synthesized by fruit are also closely related to the external growth conditions [45]. Temperature, light, and moisture all affect the contents of VOCs in the fruit [46,47]. The contents of C6/C9 aldehydes/alcohols in fruits can be increased by adjusting the temperature, light, and relative humidity [48]. Hence, attention also should be paid to these management details in actual production when aiming to improve the taste and quality of cucumbers.

## 5. Conclusions

In this study, we selected three representative cucumbers with distinct flavors (better-tasting cucumber YX, Northern China-type cucumber KX, and pickled-type cucumber GX) from 30 cucumber cultivars based on flavor-quality evaluation. The volatile compounds of fresh fruits of these three cultivars were determined by automatic headspace solid-phase microextraction coupled with gas chromatography–mass spectrometry. A total of 163 VOCs were detected in 18 cucumber flesh samples, which consisted of 13 kinds of compounds, including amines, alcohols, aromatics, phenols, sulfur compounds, halogenated hydrocarbons, aldehydes, acids, terpenoids, hydrocarbons, ketones, heterocyclic compounds, and esters, where the contents of YX were higher than those of KX and GX. In our research, heterocyclic compounds, hydrocarbons, and aldehydes were the most abundant detected compounds in cluster analysis. Through difference analysis, a total of 43 differential metabolites were selected, of which 38 were detected, and the content of YX was higher than that of GX and KX. Only three differential metabolites were identified in all three groups, and the contents were still the highest in YX, namely, alpha-cadinol, *N*-(2,3,4-trifluorobenzoyl)-l-alanine-methyl ester, and [3*S*-(3-alpha,5-alpha,8-alpha.)]-5-azulenemethanol,1,2,3,4,5,6,7,8-octahydro-alpha,alpha,3,8-tetramethyl-acetate. These three compounds have not been reported before. The results of this study are helpful in guiding the identification of VOCs and quality modification of cucumber cultivars.

## Figures and Tables

**Figure 1 foods-11-01101-f001:**
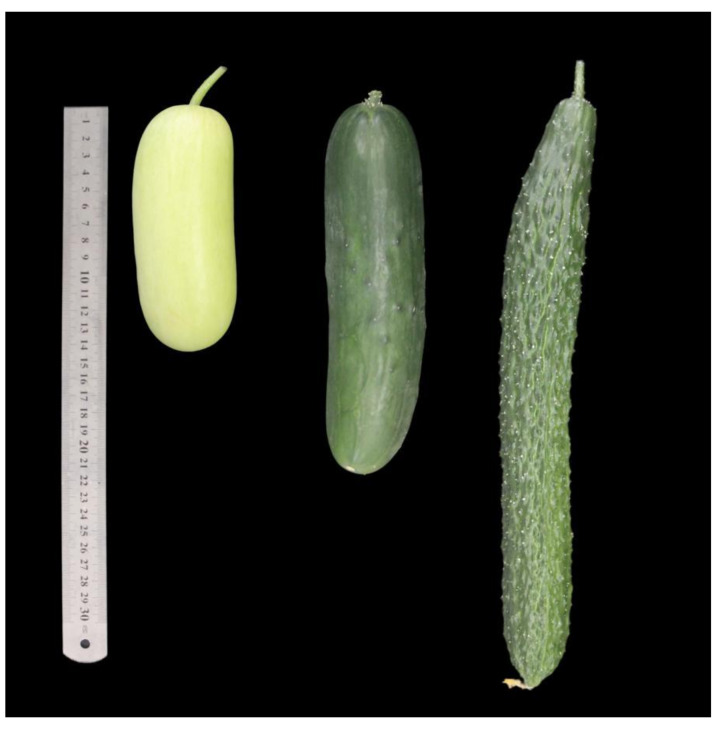
Image of the three cucumber cultivars selected for HS-SPME/GC–MS analysis. From left to right: YX, GX, and KX.

**Figure 2 foods-11-01101-f002:**
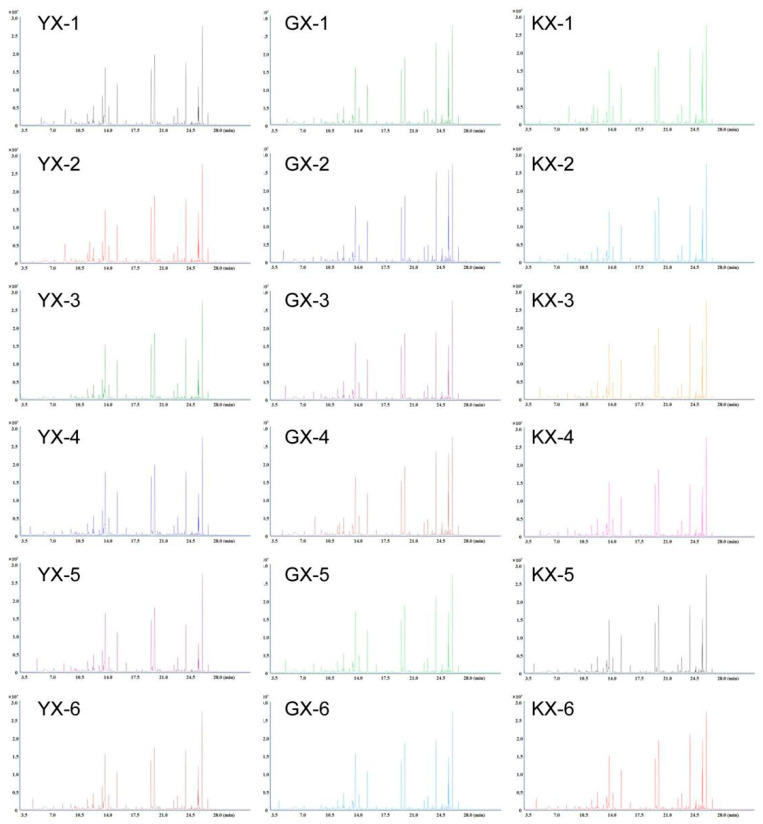
Total ion current chromatograms of volatile compounds in cucumber fruit. The abscissa is the retention time (Rt) of metabolite detection, and the ordinate is the ion flow intensity of ion detection (intensity unit: CPS, count per second).

**Figure 3 foods-11-01101-f003:**
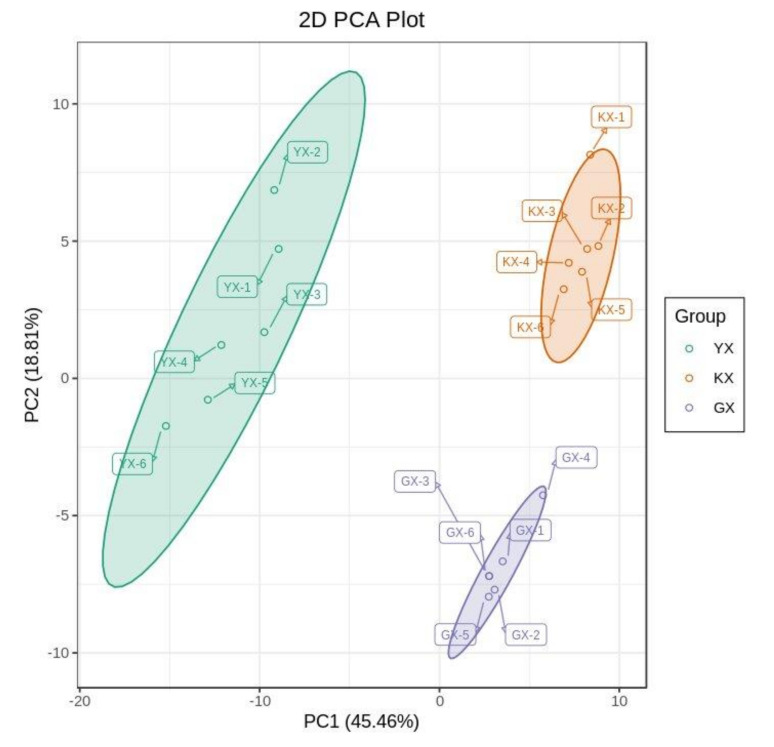
Principal components (PC) analysis of mass spectrum data of three cucumber cultivars YX, KX, and GX.

**Figure 4 foods-11-01101-f004:**
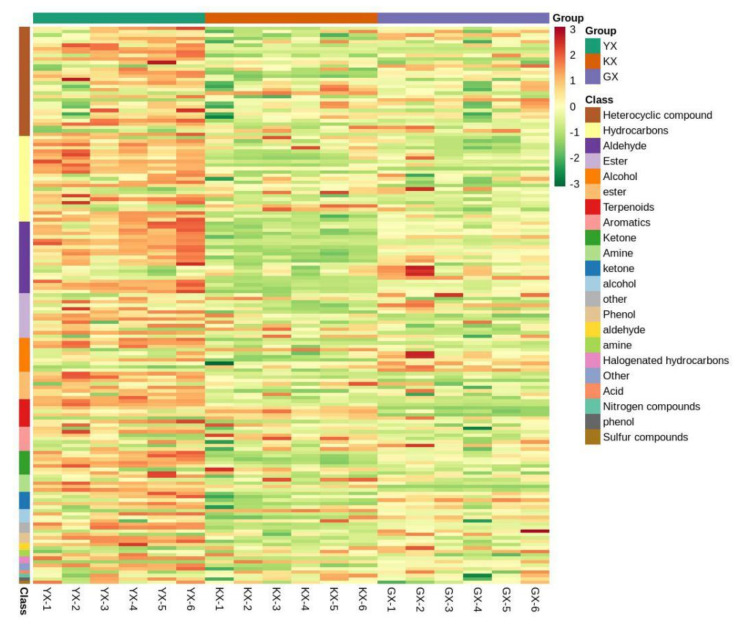
Overall clustering heatmap of samples of three cucumber cultivars YX, KX and GX.

**Figure 5 foods-11-01101-f005:**
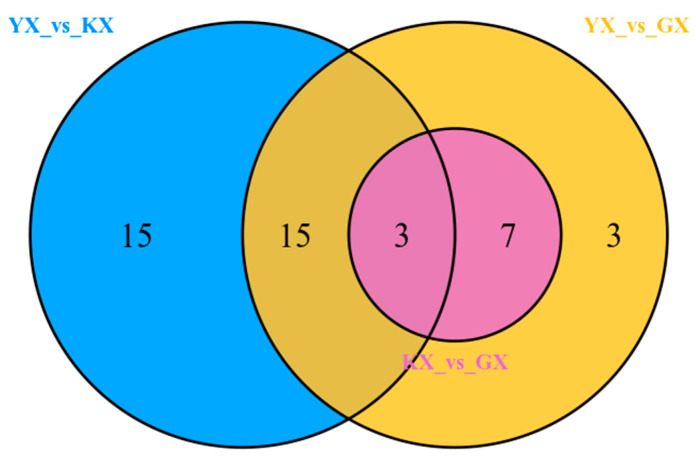
Venn diagram of differential metabolites in comparative analysis of three groups. YX_vs_KX represents those differential metabolites between YX and KX; YX_vs_GX represents those differential metabolites between YX and GX; KX_vs_GX represents those differential metabolites between KX and GX.

**Figure 6 foods-11-01101-f006:**
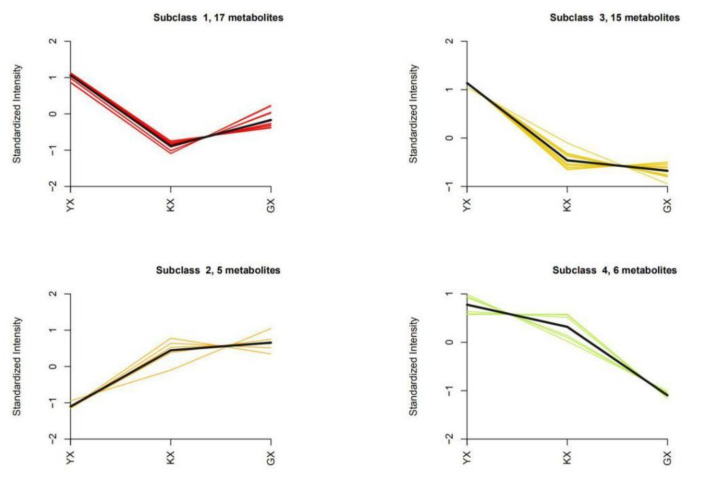
K-means map of four subclasses of differential metabolites.

**Table 1 foods-11-01101-t001:** Flavor quality evaluation of 30 cucumber cultivars.

#.	Cultivar	Fresh Cucumber-Like Flavors	Sweetness	Astringency	Total Soluble Solids (%)
1	YX	5.25 ± 0.83	3.4 ± 0.54	0.96 ± 0.36	5.2 ± 0.25
2	YN	5.13 ± 0.51	3.2 ± 1.76	1.05 ± 0.32	5.1 ± 0.47
3	C17	4.85 ± 1.16	3.15 ± 0.57	1.15 ± 1.13	5.1 ± 0.26
4	JM83	5.1 ± 2.54	3.25 ± 0.86	1.32 ± 0.88	5 ± 0.36
5	Baixiu	5.15 ± 1.56	3.2 ± 0.47	1.12 ± 0.48	5.08 ± 0.58
6	C7	4.78 ± 1.7	2.95 ± 1.12	1.4 ± 0.67	4.9 ± 0.48
7	Haiyang	5.15 ± 1.23	3.1 ± 1.04	1.25 ± 2.21	5 ± 0.95
8	Jin5-508	4.32 ± 0.78	3.11 ± 1.37	1.6 ± 1.36	4.37 ± 0.67
9	Deruite1	4.98 ± 1.28	3.1 ± 2.11	1.3 ± 1.09	4.95 ± 0.57
10	Baipi	5.05 ± 0.87	3.18 ± 1.23	1.15 ± 1.16	5.12 ± 0.79
11	Zhexiu	4.72 ± 1.28	2.8 ± 2.15	1.5 ± 1.27	4.95 ± 0.87
12	Pingwang	4.7 ± 1.36	2.84 ± 0.86	1.45 ± 1.57	4.9 ± 0.37
13	Jingtong	5.12 ± 0.76	3.2 ± 1.29	1 ± 0.27	5 ± 0.18
14	YL	4.56 ± 0.93	2.92 ± 1.82	2.1 ± 0.79	4.6 ± 0.66
15	CFCY	4.67 ± 2.21	2.75 ± 1.67	2.32 ± 2.54	4.5 ± 0.37
16	KX	4.62 ± 1.66	2.83 ± 1.52	1.18 ± 0.69	4.35 ± 0.08
17	ZN12	4.51 ± 2.18	3.51 ± 0.75	1.7 ± 2.68	4.5 ± 0.78
18	KP2	4.6 ± 2.23	2.32 ± 1.08	1.84 ± 0.95	4.17 ± 0.38
19	Xinyan	3.88 ± 0.91	3.05 ± 1.67	1.36 ± 1.78	4.95 ± 0.67
20	9930	4.45 ± 1.78	2.52 ± 1.39	1.26 ± 2.37	4.06 ± 0.46
21	Gy14	3.55 ± 1.89	2.52 ± 2.56	1.1 ± 1.09	4.12 ± 0.53
22	YL	4.56 ± 1.36	2.92 ± 0.92	2.1 ± 2.39	4.6 ± 0.48
23	GX	3.4 ± 1.63	2.1 ± 1.17	1.3 ± 1.73	3.9 ± 0.37
24	JC4	4.07 ± 1.89	2.18 ± 1.36	2.05 ± 1.66	4.2 ± 0.43
25	JY6	4.4 ± 2.1	2.49 ± 0.47	1.15 ± 2.15	3.96 ± 0.53
26	XF	4.8 ± 1.78	2.65 ± 1.89	2.25 ± 1.67	4.5 ± 0.27
27	JZ2	3.5 ± 1.89	2.8 ± 2.84	1.5 ± 2.24	4.11 ± 0.43
28	ZN18	4.32 ± 1.27	2.75 ± 1.71	1.18 ± 1.82	4.41 ± 0.33
29	JY118	4.1 ± 1.49	2.11 ± 0.74	1.5 ± 2.06	3.8 ± 0.31
30	JY30	4.45 ± 1.22	2.83 ± 0.79	1.18 ± 1.71	4.35 ± 0.51

**Table 2 foods-11-01101-t002:** K-means clustering metabolite information.

SC ^a^	Compounds	Classification	YX	KX	GX
1	Hexanal	Aldehyde	207, 381	95, 710	153, 429
1	(*Z*)-6-Nonenal	Aldehyde	221, 623	82, 261	124, 408
1	1-Hepten-3-one	Ketone	34, 832	14, 186	19, 223
1	3-Octen-2-one	Ketone	53, 015	26, 222	40, 003
1	3,5-Octadien-2-one	Ketone	1, 138, 250	541, 297	861, 772
1	Eicosane	Hydrocarbons	74, 188	31, 574	44, 022
1	Heneicosane	Hydrocarbons	503, 109	245, 410	415, 171
1	Pentadecane, 2,6,10-trimethyl-	Hydrocarbons	738, 487	226, 650	424, 901
1	3,5,5-Trimethyl-2- hexene	Hydrocarbons	15, 794	7, 245	9, 498
1	1-Ethyl-6-ethylidene- cyclohexene	Hydrocarbons	26, 013	12, 437	15, 118
1	Dimethylphosphinic fluoride	Other	39, 357	18, 034	26, 355
1	1-Chloro-3-diethylboryloxy-2,2-dimethyl-propane	Other	9, 196	2, 464	4, 232
1	1-Bromo-3-methyl- cyclohexane	Other	11, 414	4, 467	6, 260
1	2-Bromo-1,1,3-trimethyl- cyclopropane	Halogenated hydrocarbon	13, 685	2, 787	4, 834
1	Hexadecanoic acid, butyl ester	Ester	4, 291	2, 136	3, 612
1	4-Hexyn-3-ol	Alcohol	85, 161	30, 715	46, 654
1	2,4-Diamino-6-methyl-1,3,5- triazine	Heterocyclic compound	32, 258	12, 697	16, 842
2	1-Amino-4,4-dimethyl-1-(3- pyridyl)-pent-1-en-3-one	Heterocyclic compound	3, 420	7, 124	12, 151
2	1,4-Dimethyl-2,3-diazabicyclo [2.2.1]hept-2-ene	Heterocyclic compound	3, 462	7, 426	8, 387
2	Phytol, acetate	Alcohol	39, 578	91, 293	94, 381
2	3,7,11,15-Tetramethyl-2- hexadecen-1-ol	Terpenoid	33, 424	72, 255	69, 560
2	1-Iodo-hexadecane	Halogenated hydrocarbon	1	14, 185	10, 972
3	2-Nonenal, (*E*)-	Aldehyde	294, 342	70, 452	79, 612
3	2,6-Nonadienal, (*E*,*Z*)-	Aldehyde	2, 703, 944	957, 611	1, 013, 669
3	Pentadecane	Hydrocarbon	301, 783	146, 788	122, 120
3	1,7-Nonadiene, 4,8-dimethyl-	Hydrocarbon	118, 356	30, 569	37, 015
3	Bicyclo(3.3.1)non-2-ene	Hydrocarbon	147, 340	72, 365	69, 739
3	1-Pyridineethanamine, beta.-(2-furanyl)hexahydro-	Heterocyclic compound	11, 673	5, 048	3, 241
3	(5*R*,8aR)-5- Propyloctahydroindolizine	Heterocyclic compound	28, 341	4, 739	4, 565
3	Pyrimido[1,6-a]indole, 1,2,3,4-tetrahydro-2,5-dimethyl-	Heterocyclic compound	28, 775	15, 547	5, 831
3	2-n-Butyl furan	Heterocyclic compound	23, 856	12, 474	11, 187
3	2,2′,5,5′-Tetrahydro-2,2′-bifuran	Heterocyclic compound	146, 740	29, 805	40, 074
3	Caryophyllene oxide	Terpenoid	52, 982	22, 729	21, 174
3	alpha-Cadinol	Terpenoid	28, 669	8, 525	2, 101
3	l-Alanine, *N*-(2,3,4-trifluorobenzoyl)-, methyl ester	Ester	30, 477	9, 536	4, 179
3	5-Azulenemethanol, 1,2,3,4,5,6,7,8-octahydro-alpha, alpha,3,8-tetramethyl-acetate, [3*S*-(3-alpha,5-alpha,8-alpha)]-	Ester	31, 660	10, 633	3, 405
3	1-Phenylcyclohexylamine	Amine	37, 419	20, 625	15, 406
4	Bicyclo[5.2.0]nonane, 2-methylene-4,8,8-trimethyl-4- vinyl-	Terpenoid	35, 810	35, 886	4, 460
4	1,4,7-Cycloundecatriene, 1,5,9,9-tetramethyl-, *Z*,*Z*,*Z*-	Hydrocarbon	311, 884	181, 323	42, 570
4	1-Hexen, 2-(p-anisyl)-5-methyl-	Hydrocarbon	17, 985	12, 749	5, 681
4	Cyclobutanecarboxamide, *N*-(3-methylphenyl)	Amine	25, 999	25, 703	1, 856
4	4-(Benzyl-ethyl-amino)-butyric acid, methyl ester	Ester	12, 004	8, 754	3, 792
4	3a,7-Methano-3aH- cyclopentacyclooctene, 1,4,5,6,7,8,9,9a-octahydro-1,1,7- trimethyl-, [3aR-(3a-alpha,7-alpha,9a-beta)]-	Terpenoid	17, 788	16, 926	5, 272

^a^ Subclass, corresponding to the subclass category number in the k-means diagram. Shown under YX, KX, and GX are the relative contents of differential metabolites according to the indicated cultivar.

## Data Availability

Data a contained within the article or Appendix A.

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
