# Peer review of "Characterization of Differences in the Composition and Content of Volatile Compounds in Cucumber Fruit"

_foods, 2022, doi:10.3390/foods11081101_

Round 1

Reviewer 1 Report

  • the first sentence of abstract is no properly constructed, in terms of its language and gramma,
  • first paragraph of the introduction is full of grama errors
  • l. 36 is not correct, not all volatiles are responsible for flavour, only small part of them!
  • l. “were planted” is not correct form
  • storage in -80C might affect the volatile composition significantly as was indicated before, this information should be incorporated in the manuscript
  • l. 87 change the sentence please
  • l. 103 Extraction head ?? what does it mean, it has to be change
  • l. 111 SPME
  • l.112, there is two “and”, change it please
  • What does it mean “clear flavor”?
  • How authors identified components? 163 compounds identified by 1-D GC-MS is a lot ..
  • How the identity of the components was confirmed? not retention indexes were presented or determined
  • I don't see relation between the sensory analysis experiments and volatiles analysis, volatiles do not affect the chosen taste descriptors
  • In my opinion "the story" of the manuscript was not properly design, there is no clear objective of the research as well as lack of clear conclusion
  • the language must be checked carefully, there is too many mistakes to point out them all
  • Not all of these volatiles affect flavour, the message send by this article is confusing

Author Response

Comments and Suggestions for Authors

the first sentence of abstract is no properly constructed, in terms of its language and gramma, first paragraph of the introduction is full of grama errors

Reply:We have scanned the entire manuscript and tried our best to rephrase the relevant sentences in this revision. The first paragraph of the introduction was revised accordingly.

l.36 is not correct, not all volatiles are responsible for flavour, only small part of them!

Reply:Thanks for pointing this out. The sentence was revised to “Volatile organic compounds (VOCs) contribute to fruit flavor”.

l.“were planted” is not correct form

Reply:Thanks for pointing this out. The sentence was revised to “Ten plants of each cultivar were grown in a greenhouse”.

storage in -80C might affect the volatile composition significantly as was indicated before, this information should be incorporated in the manuscript

Reply: We agree with reviewer’s assessment about the effect of storage temperature on volatile composition. In our study, the the cucumber fruit samples were storage in -80 ℃ for 6 days before HS-SPME/GC-MS analysis. We incorporated this information in the discussion section.

l.87 change the sentence please

Reply:We thank the reviewer for the advice. The sentence was revised to “Sensory evaluation was conducted by 10 trained graduate students with five men and five women, aged 21–25”.

l.103 Extraction head ?? what does it mean, it has to be change

Reply:We are sorry for the mistake. “head” was revised to “fiber”.

l.111 SPME

Reply:Typo corrected.

l.112, there is two and, change it please

Reply: Yes, the sentence was revised to “The samples were removed from the -80 ℃ freezer and ground into a fine powder with liquid nitrogen.”

What does it mean clear flavor?

Reply: We are sorry for the confusing. It should be fresh cucumber-like flavours.

How authors identified components? 163 compounds identified by 1-D GC-MS is a lot ..

Reply: The mass spectrum was matched with the self-built MWGC database (Metware Biotechnology Co., Ltd. Wuhan, China) or the standard NIST library.

How the identity of the components was confirmed? not retention indexes were presented or determined

Reply: We are sorry for the confusing. Identification of volatile compounds was achieved by matching the mass spectra with the self-built MWGC database (Metware Biotechnology Co., Ltd. Wuhan, China) or NIST library and based on the linear retention indexes of compounds. Retention indexes (RI) and retention time were provided in Table S1.

I don't see relation between the sensory analysis experiments and volatiles analysis, volatiles do not affect the chosen taste descriptors

Reply: Thanks for pointing this out. In this manuscript, we used sensory analysis experiments to evaluate the odor, taste and mouthfeel to arrive at an overall impression of the 30 cucumber cultivars. Then, 3 representative cultivars, YX, KX and GX were selected for volatiles analysis, to identify compounds in cucumbers that could be contributing to off aromas desirable flavors. The relation between chemical and sensory is an extensive research area.

In my opinion "the story" of the manuscript was not properly design, there is no clear objective of the research as well as lack of clear conclusion. the language must be checked carefully, there is too many mistakes to point out them all. Not all of these volatiles affect flavour, the message send by this article is confusing

Reply: The aim of this work is to study the impact of cultivars on the concentration and the combination of VOCs in cucumber which could be integrated into breeding programs for future flavor improvement. We have scanned the entire manuscript and tried our best to address the issues and deficiencies that were pointed out, and revised the manuscript accordingly. We hope that these revisions improve the manuscript such that you and the reviewers now deem it worthy of publication in FOODS.

Reviewer 2 Report

In my opinion, the originality of the manuscript presented for evaluation is relatively low. I do not agree with the authors that the scope of knowledge regarding the identification of volatile compounds in cucumber is fragmentary. In databases I found nearly 40,000 different articles, the earliest of which was dated 1974. Nevertheless, demonstrating the significance of variability or differences in the profile of volatile compounds is important and interesting from the point of view of selecting the appropriate organoleptic characteristics of the raw material for industrial and consumption purposes. For this aspect, I estimate the great application potential of the obtained results.

In my opinion, the work needs to be improved due to the following comments.
The introduction requires rewriting. In my opinion, the authors did not fully precisely define the purpose of the work, and thus the potential application aspect of the received data. There is also too little information about the raw material itself, for example regarding the role in human nutrition. The information from lines 47 to 57 hardly fits the introduction. I suggest moving to the discussion.

the material and methods section also needs to be completed. There is no information on how the quantitative calculations were performed: standards, calibration etc.
Table 1 is sweetness the same as sugar content. I conclud that it is not. The information given in the methods shows that the authors measured the sugars content but the results were not presented.
I also have reservations about the data presented in the supplement. In table S3 characters from the Chinese alphabet appear, which makes the results incomprehensible to the reader.

Author Response

In my opinion, the originality of the manuscript presented for evaluation is relatively low. I do not agree with the authors that the scope of knowledge regarding the identification of volatile compounds in cucumber is fragmentary. In databases I found nearly 40,000 different articles, the earliest of which was dated 1974. Nevertheless, demonstrating the significance of variability or differences in the profile of volatile compounds is important and interesting from the point of view of selecting the appropriate organoleptic characteristics of the raw material for industrial and consumption purposes. For this aspect, I estimate the great application potential of the obtained results.

Reply:Thank you very much for your careful investigation throughout the whole text. We tone down the statement by writing: However, research on the types and contents of VOCs in different cucumber cultivars remains fragmentary

In my opinion, the work needs to be improved due to the following comments.The introduction requires rewriting. In my opinion, the authors did not fully precisely define the purpose of the work, and thus the potential application aspect of the received data. There is also too little information about the raw material itself, for example regarding the role in human nutrition. The information from lines 47 to 57 hardly fits the introduction. I suggest moving to the discussion.

Reply: We deleted the information from lines 47 to 57 from the manuscript. The aim of this work is to study the impact of cultivars on the concentration and the combination of VOCs in cucumber which could be integrated into breeding programs for future flavor improvement. We also added more information about VOCs in plants.

the material and methods section also needs to be completed. There is no information on how the quantitative calculations were performed: standards, calibration etc.

Reply: Identification of volatile compounds was achieved by matching the mass spectra with the self-built MWGC database (Metware Biotechnology Co., Ltd. Wuhan, China) or NIST library and based on the linear retention indexes of compounds. To ensure the accuracy of the determined values, six replicates were analyzed for each cultivar. The original data files obtained by GC-MS analysis were extracted using MassHunter software (Agilent, USA), and the mass-to-charge ratio, retention time, and peak area of characteristic peaks were obtained and then analyzed statistically. Then, the original data retention index was calculated, single peaks were filtered, and quantitative analysis was carried out by an internal standard normalization method.

Table 1 is sweetness the same as sugar content. I conclud that it is not. The information given in the methods shows that the authors measured the sugars content but the results were not presented.

Reply: Thanks for pointing this out. We actually measured the total soluble solids content using a digital refractometer (ATAGO, Japan).

I also have reservations about the data presented in the supplement. In table S3 characters from the Chinese alphabet appear, which makes the results incomprehensible to the reader.

Reply: We are sorry for the mistake. In the revised table S3, the Chinese alphabets were deleted. We also add more details to help explain the main results to readers.

Reviewer 3 Report

Dear Editor 

After carefully reading the current manuscript, these are my observations.

In this work, the authors applied an automatic headspace solid-phase microextraction coupled with gas chromatography-mass spectrometry method in order to analyze the volatile compounds (VOCs) of three different cucumber cultivars. Furthermore, principal component and clustering analysis of VOCs were used, in order to differentiate and highlight the best cucumber cultivar according to their flavor characteristics. In general, experimentation data and their statistical analysis revealed interesting information that could connect VOCs composition with flavor characteristics of vegetables or fruits and thus finally their, consumers' acceptance. 

Nevertheless, there are some issues that should be discussed.

Lines 47-49. "Before analyzing the volatile components of fruit, fresh samples need to be properly separated and purified [10]. An important prerequisite for the accurate analysis of volatile substances is to select an appropriate sample separation method."

Be more specific. What do you mean by the bold parts of these sentences?

Line 58. "To date, 291 VOCs have been identified in melons [14],.." 

I believe it is irrelevant to the sentence followed.

Line 74. Please clarify if a period of 12 days is needed after flowering to result in full-grown cucumbers.

Line 109. n-Hexane.

Line 110. What standard?

Lines 112-120. According to the experimentation described in this paragraph, the grounded fruits, after being defrosted, were placed in a headspace bottle and kept under thermal conditions at 100 oC for 5 minutes in a water bath while afterward, the extraction head was inserted to conclude the solid-phase extraction stage. From this point of view, the extraction head will not absorb the real aroma compounds of cucumber that consumers perceive under usual room temperature conditions before they consume, but also the components of cucumber that are released under higher temperatures and consumers probably do not sense as a selection factor. Please discuss.

Lines 119-120. Be more clear. Did the extraction procedure hold 15 min under 100 oC? Also, what do you mean by the phrase ".....and the sample was analyzed for 5 min at 250 ℃ for GC–MS analysis"?

Lines 122-132. The most important part of the present work is the analysis of VOCs (both qualitatively and quantitatively), beyond any doubt. From this point of view, I was unable to find in the text how authors manage to positively identify the 163 mentioned volatile cucumber components.

Lines 161-175 and all over the tex: Greek letters or Z, E stereochemical symbols in nomenclature should be given in Italics form.

Line 175 and Fig 2. In this Figure, only one chromatogram is given.

Lines 305-306. Be more clear.

Lines 310-311. Probably irrelevant to the text. Please explain, why often in the text data-driven from the melon analysis is used herein?

Author Response

Lines 47-49. "Before analyzing the volatile components of fruit, fresh samples need to be properly separated and purified [10]. An important prerequisite for the accurate analysis of volatile substances is to select an appropriate sample separation method." Be more specific. What do you mean by the bold parts of these sentences?

Reply: We deleted this part from the manuscript. We attempted to show the advantages of HS-SPME sampling technique, which has advantages of high purity of the extract, avoidance of organic solvents and simple technical manipulation and can be used in combination with GC-MS.

Line 58. "To date, 291 VOCs have been identified in melons [14],.." I believe it is irrelevant to the sentence followed.

Reply: The sentence was deleted from the revised manuscipt.

Line 74. Please clarify if a period of 12 days is needed after flowering to result in full-grown cucumbers.

Reply: We are sorry for a mistake here. It is true that the cucumber fruits are still immature at 12 days after flowering, but cucumbers for flesh eating or pickling are typically harvested while still immature.

Line 109. n-Hexane.

Reply: typo corrected.

Line 110. What standard?

Reply: 2-methyl 3-heptanone.

Lines 112-120. According to the experimentation described in this paragraph, the grounded fruits, after being defrosted, were placed in a headspace bottle and kept under thermal conditions at 100 oC for 5 minutes in a water bath while afterward, the extraction head was inserted to conclude the solid-phase extraction stage. From this point of view, the extraction head will not absorb the real aroma compounds of cucumber that consumers perceive under usual room temperature conditions before they consume, but also the components of cucumber that are released under higher temperatures and consumers probably do not sense as a selection factor. Please discuss.

Reply: HS-SPME, in which a coated fused silica fiber is used to trap and concentrate analytes from a static or dynamic headspace process, was developed in 1993 and has experienced the strongest growth in research interest over the past decade. It is true that thermal processing may change chemical structures resulting in altered compounds, but our protocol has proved to be stable. The sealed bottle was balanced in a 100℃ water bath for 5 min to reduce sensor volatility due to environmental changes. In addition, before GC testing, samples need to be converted to a volatile and thermally stable form, because GC is used to separate the volatile and thermally stable substitutes in a sample. Further studies with other alternative testing methods are needed to eliminate the potential side effects of thermal processing during SPME sampling. The above description was also included in the discuss section.

Lines 119-120. Be more clear. Did the extraction procedure hold 15 min under 100 oC? Also, what do you mean by the phrase ".....and the sample was analyzed for 5 min at 250 ℃ for GC–MS analysis"?

Reply: After sample preparation, each vial was placed in a water bath at 100℃ for 5 min with agitation to reach an equilibrium state. The extraction fiber was inserted into the sample headspace bottle for 15 min to absorb VOCs. The fibre was conditioned prior to use by heating in the injection port of a GC system at 250 ℃ for 5 min.

Lines 122-132. The most important part of the present work is the analysis of VOCs (both qualitatively and quantitatively), beyond any doubt. From this point of view, I was unable to find in the text how authors manage to positively identify the 163 mentioned volatile cucumber components.

Reply: Identification of volatile compounds was achieved by matching the mass spectra with the self-built MWGC database (Metware Biotechnology Co., Ltd. Wuhan, China) and based on the linear retention indexes of compounds.

Lines 161-175 and all over the tex: Greek letters or Z, E stereochemical symbols in nomenclature should be given in Italics form.

Reply: We thank the reviewer for the suggestion. Changes were made as suggested.

Line 175 and Fig 2. In this Figure, only one chromatogram is given.

Reply: The chromatograms of the 18 tested samples were provided.

Lines 305-306. Be more clear.

Reply: We are sorry for the confusing. “were found to be higher in the fruit than in other parts, while their ratio is closely related to the cucumber flavor” was deleted from the manuscript.

Lines 310-311. Probably irrelevant to the text. Please explain, why often in the text data-driven from the melon analysis is used herein?

Reply: The description was revised to “ fatty acids-derived VOCs make significant contributions to tomato fruit flavor and human preferences”.

Round 2

Reviewer 1 Report

The suggestions were incorporated into the manuscript.

It might be accepted in the present form.

Reviewer 3 Report

In this revised form current manuscript can be considered for publication.